

# Effects of a natural precipitation gradient on fish and macroinvertebrate assemblages in coastal streams

Sean Kinard[1], Christopher J. Patrick[1] and Fernando Carvallo[2]

[1] Department of Biological Sciences, Virginia Institute of Marine Science, Gloucester Point, VA, United States of America
[2] Department of Life Sciences, Texas A&M Corpus Christi, Corpus Christi, TX, United States of America

## ABSTRACT

Anthropogenic climate change is expected to increase the aridity of many regions of the world. Surface water ecosystems are particularly vulnerable to changes in the water-cycle and may suffer adverse impacts in affected regions. To enhance our understanding of how freshwater communities will respond to predicted shifts in water-cycle dynamics, we employed a space for time approach along a natural precipitation gradient on the Texas Coastal Prairie. In the spring of 2017, we conducted surveys of 10 USGS-gauged, wadeable streams spanning a semi-arid to sub-humid rainfall gradient; we measured nutrients, water chemistry, habitat characteristics, benthic macroinvertebrates, and fish communities. Fish diversity correlated positively with precipitation and was negatively correlated with conductivity. Macroinvertebrate diversity peaked within the middle of the gradient. Semi-arid fish and invertebrate communities were dominated by euryhaline and live-bearing taxa. Sub-humid communities contained environmentally sensitive trichopterans and ephemeropterans as well as a variety of predatory fish which may impose top-down controls on primary consumers. These results warn that aridification coincides with the loss of competitive and environmentally sensitive taxa which could yield less desirable community states.

## INTRODUCTION

A warming climate warrants a better understanding of the processes that link biological communities to long-term trends in temperature and precipitation (*Wrona et al., 2006*; *Miranda, Coppola & Boxrucker, 2020*). The direct ecological effects of changes in temperature have received greater attention in the literature, but rising temperatures are also expected to alter patterns of precipitation and evaporation. A warmer, more energetic atmosphere intensifies the hydrological cycle (patterns of precipitation and evaporation), causing wet regions to become wetter and dry regions become drier (*Allen & Ingram, 2002*), as well as increasing the frequency and intensity of extreme weather events (*Held & Soden, 2006*). In general, this raises concern for freshwater ecosystems which are highly sensitive to changes in water availability and contain many species with limited dispersal capabilities (*Woodward, Perkins & Brown, 2010*).

Corresponding author
Sean Kinard, skkinard@vims.edu

Stream ecosystems are shaped by flow regimes which regulate the physical extent of aquatic habitats, water quality, sourcing and exchange rates of material, and habitat connectivity (*Rolls, Leigh & Sheldon, 2012*). Aridification increases the prevalence of droughts and flash floods which disturb local communities by imposing intolerable conditions or physically displacing individuals. Lengthening dry periods cause changes in macroinvertebrate communities where drought sensitive taxa (Ephemeroptera, Plecoptera, and Trichoptera) are replaced by drought tolerant species (*Storey, 2016*). In contrast, humid precipitation regimes have low interannual variability and frequent bank flooding that promotes hydrological connectivity and resource exchange between aquatic and terrestrial systems. Fish communities become increasingly diverse with precipitation and temperature along continental climate gradients (*Griffiths, McGonigle & Quinn, 2014*). The expansion of semi-arid regions (*Huang et al., 2016*) warrants an improved understanding of the mechanistic links between precipitation, flow regime, and aquatic biota to manage for the increasing societal demands for freshwater resources.

Hierarchical community assembly models can help us organize our hypotheses regarding impacts of climate change on stream communities (*Poff, 1997*). Assuming organisms can disperse to a habitat, they must be able to survive in the local environment (abiotic filters) and successfully reproduce in the presence of other organisms exerting pressures (biotic interactions) such as competition and predation (*Patrick & Swan, 2011*). Species have physiological tolerances (temperature, toxin concentrations, and salinity, *etc.*) which limit their distribution across environmental gradients (*Whittaker, Willis & Field, 2001*). If climate change alters those gradients, we can expect concordant changes in species distributions. However, understanding how the environment affects biotic interactions is more challenging due to the complex sets of interactions that govern these processes (*Seabra et al., 2015*).

Observational surveys of existing communities spatially distributed along environmental gradients can be used in a space-for-time substitution to infer how communities will change through time as environmental conditions shift. The approach allows for links to be drawn between climate drivers, local environmental conditions, and organism abundances. Species co-occurrence patterns along environmental gradients can also shed light on possible shifts in biotic interactions (*D'Amen et al., 2018*). However, the space-for-time approach assumes that observed ecological differences along the spatial gradient are the sole product of corresponding changes in climate. This assumption may be unfair given that biogeographical studies have revealed that dispersal limitation, habitat heterogeneity, and local evolution can also contribute to current spatial patterns in community composition (*Jacob et al., 2015*). These studies are typically large in scale, covering vast distances (thousands of km) to capture climate gradients. These large scales make the precise mechanisms for observed biological changes difficult to ascertain due to covarying environmental variables (*e.g.*, elevation, geology, human impacts). Thus, while current literature demonstrates that biome shifts occur across temperature and latitudinal gradients (*De Frenne et al., 2013*), the value of these observational studies for forecasting community responses to climate change is hindered by the many confounding variables. The power of using the space-for-time approach to delineate the intricacies of hydrologic

cycle-ecosystem relationships is enhanced in study systems with limited confounding environmental variables (temperature, elevation, distance, and underlying geology).

The Texas Coastal Prairie (TCP) within the Western Gulf coastal grasslands is an ideal system for evaluating the effect of hydrologic climate change on ecological communities. It is located within the Western Gulf coastal grasslands which are a subtropical ecotone that spans Louisiana, Texas, and northern Mexico's coastal areas. The system encompasses the sharpest non-montane precipitation gradient in the continental United States. The climate becomes more arid as you move west, with gradual change for much of the coast and a region of rapid change located in southern Texas. In this region the annual rainfall changes from 55 cm yr$^{-1}$ (semi-arid) to 135 cm yr$^{-1}$ (sub-humid) over a 300 km gradient, but there are minimal changes in elevation, air temperature, underlying geology, and human land use. The region is characterized by gently rolling landscapes (slopes < 5%), afisol soils, streams with forested riparian zones, and a widespread conversion of grasslands to the agricultural production of cattle, cotton, corn, and soy products (*Chapman, 2018*). As conditions become wetter, there is an observable ecological shift from mesquite groves in the semi-arid West to live oak and pecan forests towards the East. The TCP is an ideal study region for isolating precipitation influences on natural ecosystem processes because of the minimal impact of covarying predictors that typify climate gradient research.

Despite the intrinsic value of this region as a candidate for climate gradient research, there is limited prior biological sampling by governmental agencies of running waters in the TCP. To address this need, we conducted the first dedicated survey of streams across the climate gradient. We applied rapid bioassessment protocols to 10 USGS-gauged (U.S. Geological Survey), wadeable streams for characterization of fish, benthic macroinvertebrates, and quantification of environmental variables. Our objectives were to: (1) Isolate and clarify the effects of annual precipitation on patterns in the diversity and composition of fish and macroinvertebrates communities, and (2) specify the hydrologic and water quality predictors that mediate the effects of precipitation on community assembly. We expected that annual precipitation would be positively correlated with community diversity because humid precipitation regimes are expected to create more stable environmental conditions by creating predictable flow regimes which promote the development of greater biodiversity (*Boulton et al., 1992*; *Bunn & Arthington, 2002*). We further expected that evapotranspiration by riparian vegetation would increase solute concentrations in semi-arid streams, particularly during base flows (*Tabacchi et al., 2000*; *Lupon et al., 2016*), creating environmental filters that limit recruitment of sensitive fish and macroinvertebrates (hereafter referred to as invertebrates).

## METHODS

### Study region

The Texas Coastal Prairie contains grassland prairie with forested areas occurring primarily along riverine systems. During March and April of 2017, we sampled ten, wadeable, perennial streams which span 12 counties from Kleberg County to Montgomery in South-Central Texas, USA (Fig. 1). Each study site was located within 100 m of a USGS

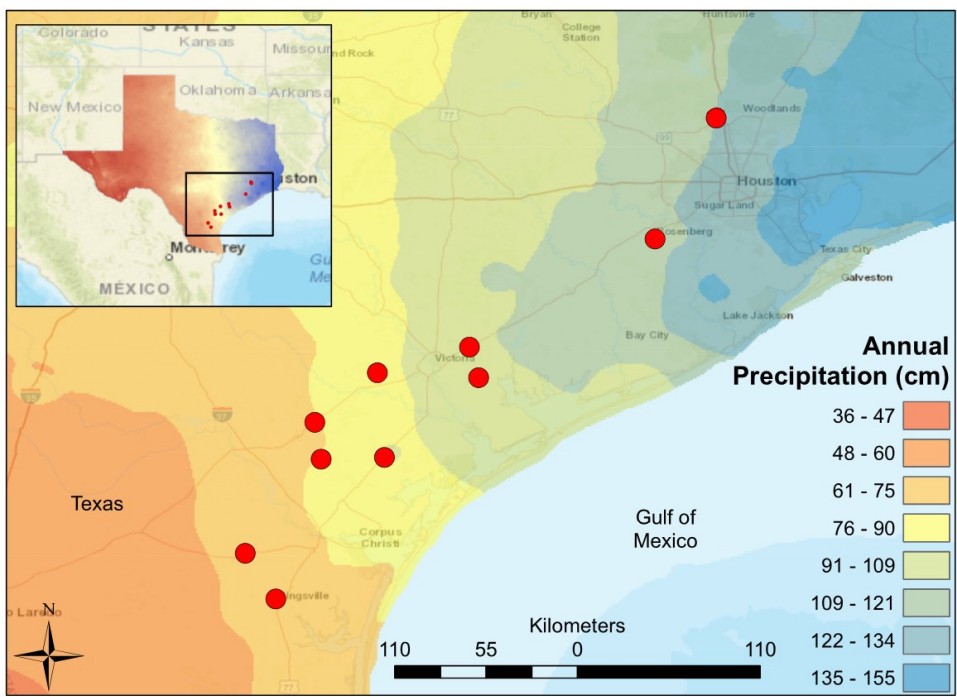

**Figure 1** **Map of South-Central Texas, where 10 USGS gaged streams were sampled in the spring of 2017.** An annual precipitation overlay indicates that the sample sites span a gradient from 61 cm/yr in the Southwest to 134 cm/yr in the Northeast.

stream gauge which continuously monitor streamflow and climate data year-round. Study sites were chosen to maximize differences in precipitation with minimal changes in underlying geology and elevation. The annual precipitation ranges from 61–121 cm within the study region which spans a linear distance from end to end of 378 km (*Falcone, 2011*). The surface geology is characterized by fine clays, quaternary and sedimentary sand. The streams have similar elevations (14–62 m), substrates (quaternary), and average air temperatures (19.8–22.1 °C) (*Falcone, 2011*). Sampling was conducted by students and faculty at Texas A&M (Corpus Christi) under permit SPR-0716-170, granted by Texas Parks and Wildlife Department.

## Biological sampling

Fish communities were sampled using a Smith-Root LR-24 Backpack in a single pass survey (*Lamberti, 2017*). Each reach length was 25 times the average stream width, in accordance with EPA rapid bioassessment protocols (*USEPA, 2019*). Low variability in stream withs (4.9 ± .6 m) resulted in comparable catch effort across sites, so fish abundances were reported in terms of catch per sample event. Fish species were field identified to species using a field guide (*Thomas et al., 2007*) and photographed. Several specimens of each species were euthanized using tricaine mesylate (MS-222) and stored in >70% denatured ethanol as voucher specimens for lab confirmation of species identification. Fish Voucher specimens were identified using the Texas Academy of Science dichotomous key (*Hubbs, Edwards*
_& Garrett, 2008_) and cross referenced with field identifications. Vertebrate sampling was permitted by the Institutional Animal Care and Use Committee, Texas A&M University Corpus Christi (AUP# 05-17).

Invertebrates were collected using a 0.305 m wide D-frame net equipped with 500-$\mu$m mesh. Twenty 0.093 m$^2$ samples were collected _via_ a combination of kick and sweep (15 s duration) sampling from a representative distribution of best available habitat (riffles, large woody debris, overhanging vegetation) (_Southerland et al., 2007_). Samples were pooled in a 500-$\mu$m sieve bucket where larger sticks and leaves were rinsed and removed. The captured invertebrates and remaining debris were preserved in 95% EtOH for transport to the lab. In the lab, samples were spread across a gridded sampling tray and randomly selected grid cells were picked to completion until the total count was >300 individuals (_USEPA, 2015_). Samples containing less than 300 individuals were picked to completion. Invertebrates were identified to lowest taxonomic resolution (typically genus) using taxonomic keys cross referenced with species observations recorded by the TCEQ's (Texas Commission on Environmental Quality) Surface Water Quality Monitoring Program (_Wiggins, 2015_; _Cummins & Merritt, 1996_). The sum of individuals in each taxon were multiplied by the fraction of unpicked sample and reported as abundance of individuals per square meter.

### Environmental data

For each stream, we averaged values for each of the following habitat measurements that were taken at 4 cross-sections spaced 25m apart. A Rosgen Index value was calculated by dividing the bank-full width by the maximum depth (_Rosgen, 2001_). Bank height was recorded as vertical difference between water level and the height of the first bench. We estimated sediment grain size within each cross-section using Wentworth size categories to calculate a median grain-size (d50) (_Wentworth, 1922_). Oxygen, temperature ($T_{water}$), conductivity, turbidity, and pH were measured at each point using a YSI ProDSS multiparameter probe. Two 60 mL water samples were collected and filtered through a pre-combusted (500 °C for 4 h) glass fiber filter (Whatman GF/F) into acid washed amber bottles, transferred to the lab in a cooler on ice, and stored frozen (−20 °C) until analysis for nutrients ($NH_4^+$, $NO_3^-$, and $PO_4^-$). Water samples were run using colorimetric methods on a latchet autoanalyzer by the Oklahoma University Soil Water and Forage Laboratory.

In addition to the habitat metrics measured in the field, we gathered climate and watershed data, from the US Geologic Surveyors Geospatial Attributes of Gages for Evaluating Streamflow, version II dataset (_Falcone, 2011_). A twenty-year continuous daily flow record was downloaded for each site (except Tranquitas Creek which only had 4 years of available data) from the USGS Water Services (https://waterservices.usgs.gov).

### Analyses

Due to a small number of sample sites and replicates, the statistical analyses relating environmental drivers to organismal responses were restricted to six _a priori_ environmental predictors. Annual precipitation was evaluated to identify gradient effects. Channel shape is a product of flow regime, slope, substrate, and bank stability and was summarized by
the Rosgen index. We included conductivity and $NH_4^+$ to evaluate water quality. Since the selected streams were deliberately chosen to be wadeable at base flow, we calculated two flow metrics to approximate the typical flow regime of each site in the context of seasonal droughts and floods, as well as overall variation in flow: Flash Index (cumulative changes in day to day daily flow/cumulative flow) and the Low-Flow Pulse Percent (LFPP = times where daily discharge drops below the 25th percentile) (*Olden & Poff, 2003*; *Patrick & Yuan, 2017*).

We used linear regression and Pearson correlation coefficients to identify potential confounding relationships between precipitation and each environmental predictor. We then, used singular value decomposition of the centered and scaled data matrix in a principal component analysis with all six environmental predictors. The environmental PCA and associated exploratory results are described in the supplemental data.

For each community (fish and invertebrate) we estimated coverage and Hill diversity metrics (*Roswell, Dushoff & Winfree, 2021*). We used coverage-based estimates of species richness, Shannon entropy and Gini-simpson index (*Chao et al., 2014*). While richness and Gini-Simpson index values are reported in supplemental materials, further analyses and discussion regarding diversity utilize the Shannon Entropy because it is not overly sensitive to either rare or common species. We used univariate regression to evaluate community diversity relationships with the precipitation gradient and each environmental predictor. We also performed exhaustive multiple regression with an additive global model utilizing all six environmental predictors and ranked them using Aikake's information criterion corrected for small sample sizes ($AIC_c$). All the results were compared to the best overall model by calculating the difference in $AIC_c$ values ($\Delta AIC_c$). Models with $\Delta AIC_c < 2$ were considered to have substantial support (*Burnham & Anderson, 2002*). Diversity Hill metrics were calculated using the iNEXT package (*Hsieh, Ma & Chao, 2020*) in R (*R CoreTeam, 2018*).

To discern compositional shifts in fish and invertebrates across the precipitation gradient, we used Redundancy Analysis (RDA) on Hellinger-transformed community data, constrained to the six environmental variables in Table 1 (*Legendre & Gallagher, 2001*; *Legendre & Legendre, 2012*). We then fit vectors to the species and environmental variables where the direction of each arrow is determined by the average directional cosines from the origin to site values within the ordination. Significant vectors had an associated *p*-value <0.05. Ordination and vectors were calculated using the 'rda' and 'envfit' functions respectively in the vegan package in R (*Oksanen et al., 2019*; *Bellier, Borcard & Legendre, 2012*). Statistics and analytical R scripts for analyses described above are reported in the Supplemental Information.

## RESULTS

### Fish community

Eighteen fish species were identified. Proceeding from semi-arid to sub-humid sites, Shannon entropy increased from 1.6 to 6.1 and richness increased from 2 to 9 species (Fig. 2). Univariate regressions indicate that Shannon diversity is positively correlated with

**Table 1** Univariate regression summary statistics and multiple regression relationships predicting fish and invertebrate Shannon entropy using environmental predictors.

| Response | Input | Slope | $R^2$ | F-stat | df | p-value | Multiple regression |
|---|---|---|---|---|---|---|---|
| $Shannon_{Fish}$ | Precipitation | 0.056 | 0.576 | 10.885 | 2 | 0.011[*] | +[*] |
| $Shannon_{Fish}$ | Flashiness | 1.430 | 0.038 | 0.316 | 2 | 0.589 | + |
| $Shannon_{Fish}$ | Channel shape | 0.113 | 0.069 | 0.597 | 2 | 0.462 | + |
| $Shannon_{Fish}$ | Low flow pulse % | −0.083 | 0.222 | 2.281 | 2 | 0.169 | − |
| $Shannon_{Fish}$ | $NH_4^+$ | −13.221 | 0.326 | 3.877 | 2 | 0.084 | − |
| $Shannon_{Fish}$ | Conductivity | −0.920 | 0.413 | 5.636 | 2 | 0.045[*] | − |
| $Shannon_{Invert}$ | Flashiness | 8.651 | 0.061 | 0.517 | 2 | 0.493 | + |
| $Shannon_{Invert}$ | Precipitation | −0.085 | 0.057 | 0.487 | 2 | 0.505 | − |
| $Shannon_{Invert}$ | Low flow pulse % | −0.489 | 0.336 | 4.056 | 2 | 0.079 | −[*] |
| $Shannon_{Invert}$ | $NH_4^+$ | −2.519 | 0.001 | 0.004 | 2 | 0.950 | − |
| $Shannon_{Invert}$ | Conductivity | −0.884 | 0.017 | 0.135 | 2 | 0.723 | − |
| $Shannon_{Invert}$ | Channel shape | −0.899 | 0.193 | 1.917 | 2 | 0.204 | − |

**Notes.**
[*]Denotes a *p*-value < 0.05 or an Δ AICc < 2.

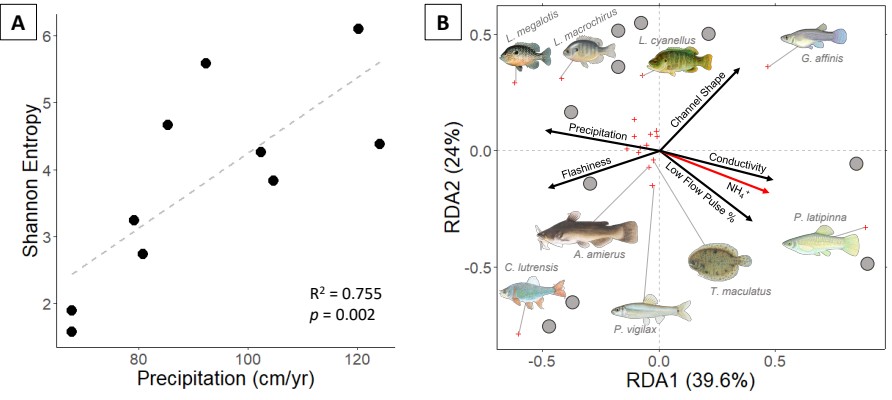

**Figure 2** **(A) Fish Shannon-Hill diversity plotted against annual precipitation (cm/yr); (B) fish community ordination using Hellinger transformation and redundancy analysis. Axes labels display the proportion of the variance explained as a percentage.** Sites are represented by grey circles. Species are represented by red crosses, with species labels and reference images added to outer members. Environmental variables are shown in arrows and the significant ones are presented in red.

precipitation and negatively correlated with conductivity (Table 1). Multiple regression indicates that precipitation alone is the strongest predictor of Shannon diversity.

Communities along the precipitation gradient are stratified in ordination space with significant vector loading on $NH_4^+$ (Fig. 3). The position of species and sites indicate that compositions shift from small, elongate live-bearer taxa (*Poecilia latipinna*, and *Gambusia affinis*) in the most arid sites to deep-bodied, nesting centrarchids (*Lepomis megalotis* and *Lepomis macrochirus*) in the more humid sites. Some mesic and humid stream communities are distinguished by the presence of *Cyprinella lutrensis*, a small, invasive habitat-generalist.

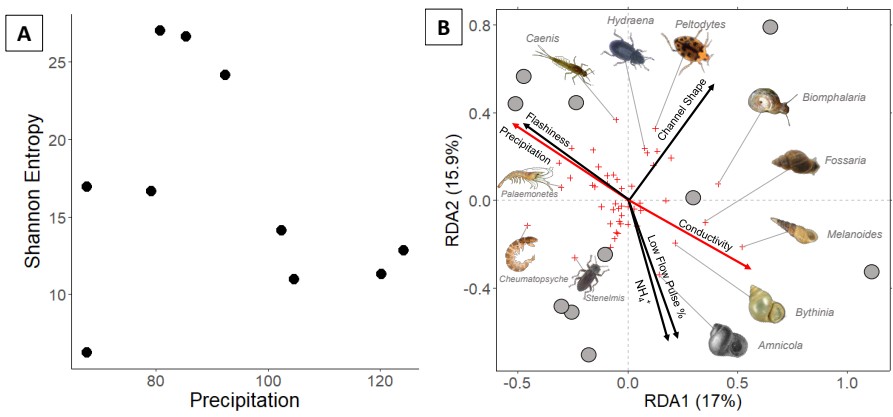

**Figure 3** **(A) Invertebrate diversity plotted against annual precipitation (cm/yr); (B) invertebrate community ordination using Hellinger transformation and redundancy analysis.** Axes labels display the proportion of the variance explained as a percentage. Sites are represented by grey circles. Species are represented by red crosses, with species labels and reference images added to outer members. Environmental variables are shown in arrows and the significant ones are presented in red.

Species found in small numbers or at singular sites fail to produce significant vectors in the RDA. Unique species found in sites on the humid side of the climate gradient include hogchoker (*Trinectes maculatus*), black bullhead catfish (*Ameirus melas*), and blacktail shiner (*Cyprinella venusta*).

## Invertebrate community

In total, 94 invertebrate genera were identified. Invertebrate richness ranged 7–29 genera with the highest values occurring at three sites in the middle of the precipitation gradient (Fig. 3). Regression analysis indicates that Shannon entropy does not significantly correlate with precipitation. However, multiple regression indicates that invertebrate diversity relates negatively with LFPP (Table 1).

Communities along the precipitation gradient stratify in ordination space along opposite/parallel axes of precipitation and conductivity. Arid communities are strongly correlated with gastropods including a non-native burrowing snail (*Melanoides tuberculata*). Mesic invertebrate communities are strongly correlated with crawling beetles (*Hydraena*) and swimming beetles (*Peltodytes*). The most humid communties correlate with several Ephemeroptera (*Caenis* and *Plauditus*), Crustacea (*Palaemonetes*), Amphipoda (*Hyalella*), and Trichoptera (*Cheumatopsyche*).

## DISCUSSION

Our goal was to quantify patterns in the diversity and composition of stream communities along an extreme precipitation gradient to better understand how streams might respond to future changes in mean annual rainfall. We identified compositional shifts in both fish and invertebrate communities along the precipitation gradient. We also observed a positive relationship between fish diversity and mean annual rainfall, matching expectations, whereas invertebrate diversity did not exhibit the expected relationships with rainfall.

Changes in water solute concentrations and flow regime appear to be additional important drivers of community responses.

The fish communities displayed increasing diversity moving from the drier to wetter sides of the survey region. Fish diversity increased with precipitation but was negatively related to conductivity, and $NH_4^+$. Elevated conductivity and $NH_4^+$ in semi-arid streams exhibited levels similar to urbanized streams (*Hatt et al., 2004*), creating stressful osmotic and toxic conditions for fish (*Redding & Schreck, 1983*; *Lock & Bonga, 1991*). Elevated $NH_4^+$ has been shown to be directly toxic to many fish (*Randall & Tsui, 2002*) and has also fueled cytotoxic algal growth (*Fetscher et al., 2015*). Elevated solute concentrations were likely driven by evaporation, the watershed area/discharge ratio, and the greater influence of wastewater effluent on low discharge streams that typify semi-arid streams (*Williams, 1999*; *Dehedin et al., 2013*). We interpreted these patterns to mean that as conditions become drier, water quality imposes abiotic filters on fish assembly which reduce overall community diversity and selects for taxa with specialized adaptations for the harsh conditions.

Communities in semi-arid streams were composed of small, live-bearing, omnivores able to tolerate high salinities including Sailfin Molly (*Poecilia latipinna*, 95 psu) and Western Mosquitofish (*Gambusia affinis*, 58.5 psu) (*Page & Burr, 1991*). The strongest compositional shift observed were increases in the abundance of centrarchids (sunfish) with increases in annual rainfall. Centrarchid species have 3–7 year lifespans, breed annually, build nests, and are omnivores (*Cooke & Philipp, 2009*). Additional increases in diversity towards the wetter side of the climate gradient included the addition of black bullhead catfish (a demersal, nesting omnivore), and several shiner species (small broadcast spawning minnows). These organisms require conditions that are stable across years as well as suitable substrate for rearing young, suggesting that conditions in semi-arid sites were excluding these taxa through environmental filtering. Additionally, some of the sub-humid and mesic sites also had seasonally migrating taxa including Rio Grande Cichlid (*Hericthys cyanogutattus*), Hogchoker (*Trinectes maculatus*), and American Eel (*Anguilla rostrate*) (*Rehage et al., 2016*; *Koski, 1978*; *Wenner, 1978*). These were absent from semi-arid sites. Given the similar proximity to nearby reservoirs and estuaries, migratory taxa may have been excluded from streams with habitat fragmentation, approximated here by low flow pulse %, that typify semi-arid streams (*De Jong et al., 2015*).

Contrary to expectations, Red shiners (*Cyprinella lutrensis*) were absent from semi-arid sites and were only present in four mesic and sub-humid sites. In ordination space, two sites with the highest abundances of red shiner (Aransas and Placedo) separated perpendicularly from the rainfall-gradient effects and coextended with stream morphology and hydrologic flashiness indices. High abundances of red shiner were associated with shallow riffle habitats with gravel substrates which occurred at three sites throughout the gradient. This was peculiar since red shiner are considered to be a habitat generalist and rugged invasive throughout the United States (*Marsh-Matthews & Matthews, 2000*; *Matthews & Marsh-Matthews, 2007*). We suspected their apparent habitat preference was driven by competition and predation by centrarchids in nearby pool and run habitats. Although red shiners tolerate high temperatures and low oxygen, conductivity was likely excluding red shiner (salinity tolerance < 10 psu) from the arid sites (*Matthews & Hill, 1977*). In this

light, we considered hydrologic flashiness a spurious influence on red shiner distributions beyond its capacity to influence channel geomorphology.

LFPP approximated drought prevalence and was the sole significant predictor of invertebrate community diversity. In addition to LFPP, the top-ranked multiple regression models also implicated $NH_4^+$ was an effective predictor of invertebrate diversity. These results corroborate expectations for the ramping disturbance conditions typical of droughts in which water availability and quality diminish over time. Compared to fish, invertebrates have restricted in-stream mobility and traditionally seek refuge in the hyporheic zone, interstitial spaces, and in some cases utilize desiccation-resistant life-stages (*Boulton et al., 1992*; *Boulton, 2003*). Here, Semi-arid community compositions included a higher proportion of gastropods which are well adapted to the stresses that characterize increased LFPP. For example, *M. tuberculata* were the most abundant primary consumers in the semi-arid streams and can resist the osmotic stress imposed by drought conditions with a broad range of salinity tolerance (0–23 PSU). This species is also well-adapted to survive and reproduce throughout periodic dewatering due to its rapid maturation (21–62 days), asexual reproduction, and internal offspring gestation (*Farani et al., 2015*).

Despite relating with LFPP, invertebrate diversity did not correlate linearly with precipitation. Instead, invertebrate diversity peaked in the middle of the rainfall gradient. The lack of a linear correlation between invertebrate diversity and precipitation may have been caused by the inherently larger species pool for invertebrates which included more taxa with biological adaptations to drought compared to fish (*Eriksson, 1993*). The peak likely represented the transition zone where taxa common on each side of the gradient were able to co-occur. Alternatively, the driest site (Tranquitas Creek) displayed uncharacteristically low diversity compared to other semi-arid sites and may constitute an outlier. When removed, invertebrate diversity correlated negatively with precipitation ($R^2 = 0.43$, *p*-value $= 0.06$). Regardless, the relation between precipitation and invertebrate diversity remains unclear.

Invertebrate community compositional shifts with rising precipitation invite continued assessment on the following speculative premises within the region: (1) The observed shift in primary consumers from short-lived, euryhaline dipterans and gastropods to ephemeropterans and trichopterans, environmentally sensitive species with longer lifespans, pointed towards improved water quality conditions and hydrologic stability (*Rosenberg & Resh, 1993*; *Jackson & Sweeney, 1995*). Taken further, this pattern alludes towards the evolutionary trade-off between aridity tolerance and competitive specialization (*Fréjaville et al., 2018*). (2) The increased prevalence of shredder-crustaceans (amphipods and crayfish) at wetter sites pointed towards a possible shift in available basal resources; precipitation-mediated shifts in riparian vegetation from evergreen, xeric mesquite trees to deciduous hardwoods could bring about increased allochthonous inputs to support more shredder taxa (*Giling, Reich & Thompson, 2009*). (3) The decreased abundance of odonate and hempiteran centrarchids (*Dahl & Greenberg, 1998*). In this way, top-down trophic interactions at predators may have been due to competition with and predation by insectivorous sub-humid sites could have restricted invertebrate communities to species

with anti-predator adaptations including small size, passive foraging strategies, camouflage, and armoring (*Straile & Halbich, 2000*).

While this survey only consisted of 10 streams, it is the first published rapid bioassessment of systems along the rainfall gradient on the Texas Coastal Prairie. The results largely conform to *a priori* hypotheses indicating that the region represents a promising study region for climate research. In addition to its capacity for a space for time substitution, the TCP is poised to provide real-time data on the effects of climate change on ecosystems. Future research in this region would benefit from higher frequency sampling over a longer time period and quantification of invertebrate and fish functional traits. An in-depth time series study would allow for evaluation of how these communities change across seasons, how they respond to periodic droughts and floods, and how stable the communities are through time. More detailed quantification of the fish communities through depletion surveys and invertebrate communities *via* biomass cores would allow for greater characterization of the relative abundance of different taxa through time, and these could be linked to functional traits to explore the mechanisms behind some of the patterns that we observed here. A continuation of this sampling program with thorough methods will augment the analytical power, precision, and depth of this natural experiment.

Despite this study's limitations, our results highlight the breadth and far-reaching ecological consequences associated with small changes in precipitation. They warn that regions expected to become more arid, like Central and Western Texas (*Jiang & Yang, 2012*), could expect a loss of competitive taxa with low environmental tolerances as observed here with centrarchids, ephemeropterans, and trichopterans. And that in their absence, rugged and euryhaline taxa (like livebearers, burrowing gastropods and predatory invertebrates) flourish. Furthermore, this study warrants investigation to clarify the causal relationships between the ecological constraints imposed by aridity and these observed community shifts.

## ACKNOWLEDGEMENTS

Jennifer Whitt and Ian Whitt for their contributions in the field and laboratory. This work was supported by National Science Foundation grant DEB-1761677, an Early Career Gulf Research Fellowship awarded to C. Patrick, and the Texas Comprehensive Research Fund.

### Funding

This work was supported by National Science Foundation grant DEB-1761677, an Early Career Gulf Research Fellowship awarded to C. Patrick, and the Texas Comprehensive Research Fund. The funders had no role in study design, data collection and analysis, decision to publish, or preparation of the manuscript.

### Grant Disclosures

The following grant information was disclosed by the authors:

National Science Foundation: DEB-1761677.
Early Career Gulf Research Fellowship.
Texas Comprehensive Research Fund.

## Competing Interests

The authors declare there are no competing interests.

## Author Contributions

- Sean Kinard analyzed the data, prepared figures and/or tables, authored or reviewed drafts of the paper, and approved the final draft.
- Christopher J. Patrick conceived and designed the experiments, authored or reviewed drafts of the paper, and approved the final draft.
- Fernando Carvallo performed the experiments, authored or reviewed drafts of the paper, and approved the final draft.

## Animal Ethics

The following information was supplied relating to ethical approvals (i.e., approving body and any reference numbers):

The Institutional Animal Care and Use Committee, Texas A&M University Corpus Christi approved this research (IACUC Number: AUP# 05-17).

## Field Study Permissions

The following information was supplied relating to field study approvals (i.e., approving body and any reference numbers):

The Texas Parks and Wildlife Department granted field permit approval (SPR-0716-170).

## Data Availability

The fish, invertebrate, environmental data as well as the variable descriptions and the annotated scripts are available in the Supplemental Files. The analyses were conducted using R scripts.

## Supplemental Information

Supplemental information for this article can be found online at http://dx.doi.org/10.7717/peerj.12137#supplemental-information.

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
