# Peer review of "Effects of a natural precipitation gradient on fish and macroinvertebrate assemblages in coastal streams"

_PeerJ, doi:10.7717/peerj.12137_

## Round 0.1 · original submission · Major Revisions

I have now received three reviews about your manuscript, all of them from stream ecologists working with a variety of organisms. I have also read carefully the paper myself. Like all reviewers, I also believe that the manuscript has potential, but authors would have to work hard to improve the writing and almost completely re-analyse the data. Data analysis is by far the major issue with the manuscript.

The several linear regressions and the nMDS (specially the way it was interpreted) are not adequate. It's also quite confusing that the ms deals with multiple dimensions of Biodiversity, from diversity indices, species richness, and species composition without a clear integration among them. You should definitely do a better job presenting these topics in a more coherent way.

The relatively small sample size of 10 streams is also a key issue that limits the breadth of analytical approaches that can be used. As a result, I do not recommend relying on linear regressions with such a small sample size. R1 and R2 also have a similar opinion. In this context, and for the questions being asked, multivariate analysis is more reasonable. However, I have to say that using nMDS for purposes other than data visualization is not good practice, due to its several drawbacks (see also Legendre & Legendre 2012), such as instability and not correctly dealing with the mean-variance relationship (Warton et al. 2012). Besides, you didn't indicate which dissimilarity coefficient you used or if you transformed the species composition data, e.g. using Hellinger distance, as recommended (Legendre & Gallagher 2001). If you want to draw inferences from multivariate data, you definitely should use more recent model-based methods, such as those available in the gllvm and HMSC R packages. In addition to those, the package ecoCopula also does model-based ordinations for data visualization. Those are much more adequate to model mean-variance relationships.

I'll only consider a much-revised version of the manuscript, which should indicate in the rebuttal letter the response to each critique made both in the review and in the PDFs reviewers sent.

·

Basic reporting

As far as I can judge, professional English is used throughout the manuscript. However from place to place, authors have made either typographical errors (and miss words sometimes) or do no use the right tense.
li 24. why "also"?
li 43. "changes in water temperature "
li 93. "for" evaluating.
li 95. replace "to climate" by "the climate"
li 125. "was"
li 138. "were"
li 150. "field rinsed"???
li 195. "we need to know if this was "all" environmental variables or "each" environmental variables (it seems it is all.
li 219. "positively related with precipitation"
li 240. "included"
li 286. "Pisidium" in italics
li 290. Oksanen (2013) is not in the references. I guess you mean Oksanen et al. (2019)
In some place the paper is rather heavy on-going. For example the long description at li 223-230 or at li 239-250 are not very useful as they are and could be simplified. In fact li 251-260 are a direct product of what is described in li 239-250 and the two parts could be mixed and lighten.
Li 378-384. What a very, very long sentence.

Li 404-406. I hardly see the strength of this sentence. Do we need it? Either too mich or not enough. Why do not you use traits in this paper? The study would be probably benefit from this.

Li 409. Acknowledgements look strange?

Li 442. Strange references. Is it complete?

In the version I have label of Figure 1 is for Figure 2 and vice-versa? Generally speaking I find the references very "US"oriented. You may take advantage of literature abroad. May I suggest to have a look at DOI: 10.1111/j.1365-2486.2007.01375.x (sorry if I indicate a paper I contribute to. It is just an example, there are probably many others ) and also DOI: 10.1007/s10750-012-1244-4.

Raw data Are shared including the statistics, which is very nice from authors. Almost nothing is hidden and the objectives of the paper are clear but probably too ambitious with the present data set. On li 47-49 "clarifying mechanistic links between climate drivers and instream biological communities will improve our ability to predict to the effects of anthropogenic climate change on lotic ecosystems." In the present version, the approach is rather correlative than causative. This is already OK to show correlations and it is necessary to have large scale observational studies but it cannot be further argued that causal relationships are detected there.

As a result, I totally disagree with the statement on li 314-317. Correlation is not causality. By the way the observed correlation is notre straightforward. Why should precipitation influence conductivity and nutrients directly? Is it because precipitation wash of ions? In addition the next sentence is ackward as you tell us here that one of your site is biased.
I am also not convinced by you sentence on li 324-327 for the same reason. In addition could we have similar compositional shift in other environmental situations?

Experimental design

Li 158. In the appendix it is strange to see the same numbers for different taxa and different sites. Why is this?

This study is original though it would gain to add some literature that has addressed similar topic with different angles.The research question are well-defined and consist mainly in identifying patterns of diversity and composition of fish and invertebrate communities associated to changes in precipitation and identify environmental drivers that mediate the effets of climate on community processes. From the the first objective I do not see why precipitation should have a direct effet on diversity and composition. It seems that the answer lies in the second objective but it is a bit tricky and the present version is not that convincing. We would probably need some model of the cascading effects from precipitation toward aquatic communities to better catch the ideas of authors. By the way, I do not see which community processes are addressed in the present version (dispersion? reproduction)

My main concern goes to the number of sites used to address this ambitious questions. I do not think at this scale with 10 sites one can catch all the natural variability of aquatic biota even if other factors are controlled (geology, elevation, temperature).

Validity of the findings

I am quite skeptical about the validity of findings because the statistical analyses lacks strength. As I said above, authors sampled 10 sites all possible regression between pairs of environmental variables and between environmental variables and biological metrics. In the appendix-regression 125 linear regression were performed. We have another set of 52 p-values in the appendix-ordination. The chance that one p-value become <0.05 just by chance increase with the number of tests. Authors should thus use a correction to the p-values. I corrected the 125 p-values of their appendix-regression and found that from the 29 significant results, false discovery rate adjustment put to the fore only 6 pairs. I think the NMDS approach is much more useful and consistent than the single regression approach. By the way why not a multiple regression approach which would help to answer the second objective. Of course with only 10 sites, no more than 3 uncorrelated environmental drivers should be used in the model.

Additional comments

There is the general interest in using space-for-time studies to address potential environmental shifts in aquatic biota. However, in this paper, I think we need a better integration of the statistical testing and underlying hypotheses. Maybe an additional tables showing the mechanistic relationship among environmental variables could help to better discuss results and how these relationships scale up to aquatic biota would be helpful. The question remains whether 10 sites (representing a substantial amount of work) are sufficient whereas large data surveys are currently made in the US and could provide much more sites along the precipitation gradient. Maybe the NMDS thoroughly commented would be enough. Finally, the introduction addresses the environmental filers framework. It is strange that natural disturbance is not mentioned. Precipitations will change water quantity but also the extreme events will be accentuated by climate change. Natural disturbance is a very highly structuring factors of aquatic communities and dozen of papers have addressed this. This driver could be better addressed in the present paper.

·

Basic reporting

I thank the opportunity to review the manuscript by Kinard et al. In this study the authors conducted field samplings for fish and invertebrates in 10 wadeable streams along a gradient of natural precipitation. They considered it as an interesting natural experiment since other environmental variables are varying little in the region, not confounding the causal linkages between precipitation and biological changes. They found a strong compositional shift in both biological groups but only fish diversity responded positively to rainfall increase.

Overall, this is a robust study in terms of field methods, biological sampling, and processing, as well as taxonomic identification and counting. The experimental design looks indeed interesting and the results are promising. I commend the authors for the detailed biological consideration of each species found, as well as for the detailed description of biological processing. However, I have some concerns regarding the chosen analyses and mostly with the discussion. The analyses cannot fully show what the authors claim during discussion whereas the discussion itself is too focused on subjective links between species and habitat characteristics. It could be better to focus on the main point of the introduction, talking about communities in the general terms, as well as to making inferences about the future impacts of climate change.

Answering to the specific points by PeerJ:

English: English is good, but some subjective adjectives used could be avoided.

Literature: Literature is updated, but I miss some key references when discussing intermittent streams (e.g. T. Datry, N. Bonada).

Structure: Some parts of the introduction could be shorter and only included in methods. The figure must be improved and better organized in rows and columns.

Results and hypotheses: Results seem robust and well sampled, but the chosen analyses made understanding difficult.

My detailed comments can be found in the attached document that I hope can be helpful.

Regards,
Victor Saito

Experimental design

This is one of the strengths of the study. The experimental design is within a natural experiment and the samples were based on well-known protocols with large evidence of suitability to the questions made. Only small detailing is needed.

Validity of the findings

The data seem robust, but could be better presented. The NMDS is not informative on the relative abundance and fidelity of taxa to a given type of stream. I include some ways to improve it in the attached document.

Additional comments

This is an interesting study and can be a good contribution to stream ecologists. I hope you can modify it in a way that strengthens your findings and make the conclusion clearer.

Reviewer 3 ·

Basic reporting

The manuscript has an interesting focus. It is well written, but the introduction lacks context. A more robust hypothesis would make the authors' objective clearer.
More details in the PDF file.

Experimental design

There are some problems with the methods. lack of references. and the analyzes used are not the most suitable for the data. See comments and suggestions for analysis in the annexes.

Validity of the findings

At some points the conclusions are vague. they need to be more connected with the results obtained and the objectives of the study.

Additional comments

The theme of the manuscript is current and the way in which the authors justified the need to assess climate gradients in the face of climate change is very interesting. However, the data still need to be better organized to show the existing patterns. I strongly suggest rethinking some of the analyzes used. It is a robust database and contains important information about biological community changes in streams.

Annotated reviews are not available for download in order to protect the identity of reviewers who chose to remain anonymous.

---

## Round 0.2 · Major Revisions

The authors have made an excellent job addressing the previous reviewers' comments. However, there are still some minor corrections that should be incorporated into the paper in this final round of review.

Like R1, I'm not sure what you mean by "RDA constrained by precipitation". Did you mean you model Hellinger-transformed abundance data as a function of precipitation? Also, consider his/her comments on the use of Shannon entropy. Questions about diversity indices are better framed in terms of Hill numbers, since they allow you to use diversity profiles increasingly weighting rare species (see several papers by Anne Chao and the R package iNExt), differently from using a single diversity measure.

All raw data displayed in tables should be moved to the supplemental material, as well as background information on test results.

I have also made several comments in the pdf attached. Many of these comments are about data analysis and visualization, which I think should mostly be remade. Please, pay close attention to them while working on the revised version of the text.

·

Basic reporting

The study largely improved from the previous version. The authors did a good job in addressing the major points from my review. There are however some points that still need consideration (see other boxes).

1- English and writing are clear and professional.
2- Literature is appropriate.
3- The structure and figures are much better than the previous ones.
4- Results are relevant and can be used to tackle the hypotheses. Some parts are too speculative, but I detail them below.

Experimental design

The study design is much clearer now and the advantages of the study area for the hypotheses tested very well written.

1- The study is an original primary research.
2- The research question is well defined.
3- The investigation methods and analytical techniques are appropriate.
4- Methods are well described, but I miss a reference or more information for the constrained RDA.

Validity of the findings

The findings are sound in general, but I have one point of concern. The discussion around invertebrate community changes is too speculative and not based on proper results. The idea of an abrupt change is also not well based or explained. Moreover, the correlation between precipitation and diversity seen to be impacted by an outlier that could be tackled by robust regression.

1- Replication within a stream is appropriate. Replication among streams is small but the discussion around the results are appropriate and concern mostly the studied region.
2 - Necessary data was clear for this assessment.
3 - Only the discussion of invertebrates' abrupt changes are too speculative.
4 - Speculative about invertebrates are taken too far supporting the conclusion.

Additional comments

Abstract:
-change USGS gauged to ‘monitored streams’
-remove ‘significantly’
-‘a variety of omnivores and piscivores fish’
-why consider the changes abrupt?

L47: ‘in general’
L 68: Perhaps complement with 'as long as species can disperse to suitable habitats'
L 114: ‘characterization of fish and invertebrate communities and quantification of environmental variables.’
L 121: ‘The link between stable conditions and habitat heterogeneity is not very clear.’
L 147: Ok, so the sampling effort was proportional to stream width, but not equal among sites. Am I correct? How do you compare abundances among sites? Using individuals per square meter?
L 169: You need to describe this type of information for fish as well.
L 186: gathered?
L 216: Because of this property, diversity indices are commonly criticized because the changes you are seeing cannot be interpreted as pure changes in richness or equitability (e.g. Magurran and McGill's book - Measuring Biological Diversity). I would recommend you to have also a pure metric of evenness, even if only to show on the Sup. Material. This would allow us to understand the changes in Shannon Diversity.
L 233: And which function for the constrained RDA? Also, I think the specific method for constraining an RDA is new (I did not find the information on the two references provided), so a reference would be nice.
L 250: How correlated they are? Some of the Shannon diversity changes could be due to changes in evenness and this would be nice to show.
L 261: This is because it is constrained before the RDA, right?
L 277: One site with very low precipitation looks like an outlier here. You could try a robust regression that accounts for extreme values because overall the scatterplot shows a decreasing trend in diversity with increasing precipitation.
Paragraph 401: This paragraph is mostly based on non-gathered results. You do not have data on basal resources, competitive specialization, competition among species, and predation pressures. Altogether, this logical construction is too fragile. Moreover, it is not well explained why these changes could be considered as abrupt ecosystem shifts. Abrupt shifts have breaking points and your RDA shows a continuous change in composition.

Reviewer 3 ·

Basic reporting

"no comment"

Experimental design

Materials and methods:
Line 158: sweep (15 s duration) sampling from a representative distribution of best available
R: Isn't it a short time? is there any reference that can help saying that this time is enough for the collection?
Line 162-164: In the lab, samples were spread across a gridded sampling tray and randomly selected grid cells were picked to completion until the total count was > 300 individuals (USEPA 2015). Samples containing less than 300 individuals were

R: see this sentence with a different font.

R: In the end of the “Analyses” section authors can add that all scripts were available in the supplementary material.

R: In figure 3, the graphs could be enlarged. They were very good, but small.

Validity of the findings

Line 347-349: (Williams 1999; Dehedin et al. 2013). We also noted but did not quantify higher concentrations of silt in the semi-arid streams with prohibitive implications for nesting species (Jones et al. 2015).
R: I think this is speculation, maybe I would remove that sentence or identify.

R: The result of the regression between canopy cover and shannonfish remained to be discussed further. I suggest that it is not an expected result (Dala-Corte et al. 2020, Journal of Applied Ecology), but the authors need to discuss more about this because this result brings new information that is the influence of the precipitation gradient associated with canopy cover. Are precipitation and canopy cover correlated variables? It would be relevant information. Until maybe authors use the factor as a covariable. So authors can see if the canopy cover can influence the fish divesity however it depends on precipitation gradient. Also, this result “fish shannon negatively with canopy” is in the abstract ... I think that the authors can remove from abstract and discuss more about it in the discussion section.


Line 371-372: Red shiners (Cyprinella lutrensis) were curiously absent from semi-arid sites and were only present in four mesic and sub-humid sites.
R: I suggest modifying the word "curiously". Example: “Contrary to expectations Red shiners are absent….” I think the language gets more scientific.

Line 374: from the rainfall-gradient effects and coextended with Rosgen (stream morphology) and
R: I suggest removing "Rosgen" leaving only "stream morphology", since the authors have already explained that it is a proxy in the material and methods.


Line 445 -450: hey warn that regions expected to become more arid, like Central and Western Texas (Jiang and Yang 2012), could expect a loss of competitive taxa with low environmental tolerances as observed here with centrarchids, ephemeropterans, and trichopterans. And that in their absence, rugged and euryhaline taxa (like livebearers, burrowing gastropods and predatory invertebrates) flourish.
R: This part need to be summarized on the abstract it is a very important conclusion of the research.

Additional comments

Abstract:
Line 36-37: These results indicate that small future changes in precipitation regime in this region may result in abrupt transitions into new community states.
But what would these new communities look like? Modify something? I think here authors have to make it clear that the modification would not be very good as it would increase/decrease diversity? Or change competition? Function on community?

Introduction:
Line 112-119:
This study conducts the first bioassessments of stream biota in this region. So, we established a framework for discussing abiotic and biotic filtering processes in community assembly, followed up by several expectations based on existing literature and precipitation-driven patterns that can be observed at a glance within the region.
R: I understand that this is a pioneering work. But it does not justify the lack of some information. Throughout the introduction, the authors talk about changes in the biota of fish and macroinvertebrates. I think the authors can be more specific on this issue. for example: diversity increases with rainy and decreases in the dry region ... Or decreases the richness of fish species in a given place. I want to make it clear that the authors have already put references on this, I just think that adding specific information, like the one I mentioned above, would help to understand the objectives and predictions of the manuscript. In my opinion, using results from work done in drought and wetter regions would only improve the introduction of the manuscript.

References:
R: The manuscript needs a revision of the references because they are not standardized with the journal's norms.
R: For example: names are different abbreviated in the first and write out in the second.
Line 510 - Fetscher, A. Elizabeth, Meredith D. A. Howard, Rosalina Stancheva, Raphael M. Kudela, Eric D. Stein, Martha A. Sutula, Lilian B. Busse, and Robert G. Sheath. 2015. “Wadeable Streams as Widespread Sources of Benthic Cyanotoxins in California, USA.” Harmful Algae 49 (November): 105–16. https://doi.org/10.1016/j.hal.2015.09.002.

Line 514 Fréjaville, Thibaut, Albert Vilà‐Cabrera, Thomas Curt, and Christopher Carcaillet. 2018. “Aridity and Competition Drive Fire Resistance Trait Covariation in Mountain Trees.” Ecosphere 9 (12): e02493. https://doi.org/10.1002/ecs2.2493.

---

## Round 0.3 · Minor Revisions

Unfortunately the RDA ordination triplots in Fig 2 and 3 still has the precipitation plotted as a gradient and as an arrow. Please, I'd kindly ask you to implement my previous suggestions on this.

---

## Round 0.4 · accepted · Accept

Thank you for making these final changes to the figure. I'm glad to accept your manuscript as is.